# Bridging the Gap: Comprehensive Boreal Forest Complexity Mapping through LVIS Full-Waveform LiDAR, Single-Year and Time Series Landsat Imagery

**Nicolas Diaz-Kloch [1,\*] and Dennis L. Murray [2]**

[1]   Environmental and Life Sciences Graduate Program, Trent University, Peterborough, ON K9J 7B8, Canada
[2]   Department of Biology, Trent University, Peterborough, ON K9J 7B8, Canada
\*   Correspondence: nicolasdiazkloch@trentu.ca

**Abstract:** The extrapolation of forest structural attributes from LiDAR has traditionally been restricted to local or regional scales, hindering a thorough assessment of single-year versus time series predictors across expansive spatial scales. We extrapolated the vertical complexity captured by the Land, Vegetation, and Ice Sensor (LVIS) full-wave form LiDAR of boreal forests in the Alaska–Yukon–Northwest Territories region, utilizing predictors from Landsat images from 1989 to 2019. This included both single-year and long-term estimates of vegetation indices, alongside constant factors like terrain slope and location. Random forest regression models comparing the single-year and 15-year and 30-year time series models were applied. Additionally, the potential of estimating horizontal forest complexity from vertical complexity was explored using a moving window approach in the Kluane Valley. While the extended time series marginally enhanced model accuracy, a fine-tuned single-year model proved superior ($R2 = 0.84$, relative RRMSE = 8.4%). In estimating the horizontal complexity, the variance in a $5 \times 5$ moving window displayed the most promising results, aligning with traditional horizontal structure measures. Single-year Landsat models could potentially surpass time series models in predicting forest vertical complexity, with the added capability to estimate horizontal complexity using variance in a moving window approach.

**Keywords:** boreal forest; remote sensing; LiDAR extrapolation

## 1. Introduction

Forest ecosystems play a crucial role in mitigating climate change, supporting biodiversity, and providing essential resources for human livelihoods [1]. Therefore, the demand for accurate and current forest data becomes paramount in ensuring effective sustainable forest management, conserving biodiversity, and monitoring carbon sequestration and climate change effects [2,3]. Fortunately, the rise of remote sensing technologies, including satellite and airborne sensors, has revolutionized this field, providing comprehensive information across various spatial and temporal scales [4]. The implications of these advances are far-reaching, resulting in remarkable improvements in our ability to monitor and manage forest resources [5,6].

Vertical complexity is a fundamental component in affecting forest biodiversity [7]. It refers to the stratification or layering of vegetation at different heights, encompassing undergrowth, shrub layers, understory, canopy, and emergent layers. It promotes habitat heterogeneity, creating diverse niches for various organisms, and enhancing overall biodiversity and ecosystem resilience [8,9]. Therefore, it is indispensable for the conservation of endangered species and the preservation of ecosystem functions [10]. Similarly, horizontal forest vegetation, encompassing tree density, size, species composition, and gap distribution, creates heterogeneous landscapes that support diverse habitat types, species richness, and essential resources for wildlife [11,12].

With remote sensing technologies like LiDAR and satellite imagery, we can now use predictive models to accurately map and monitor forests at different spatial scales by estimating attributes like tree height, biomass, and species composition [13–18]. By combining these technologies, researchers have obtained wall-to-wall estimates of forest structure, assuming the observed relationships between LiDAR and satellite imagery hold true in areas beyond the data collection points [19]. More recently, researchers have embraced the use of time series Landsat reflectance data to better predict forest attributes [20,21]. Time series data offer a more comprehensive view of forest dynamics compared to single-year data, potentially leading to improved estimates of forest structure [22]. For example, Bolton et al. [20] showed that Landsat time series length improved accuracy over single-year data, but Bolton et al. [21] found that a fixed time extent (i.e., 30 years) does not translate to optimal predictions of forest attributes.

However, the use of satellite imagery introduces several factors that can affect data quality, sensor calibration [23], and geometric distortions [24], which thereby introduce noise and uncertainty into the data and can complicate the construction of time series predictors. In particular, cloud cover poses a challenge, as it omits information in the time series [25], requiring the use of advanced methods like spectral trend analysis to create gap-free composite imagery [26]. Similarly, when using multiple sensors or satellites with different characteristics to create a time series, ensuring data homogeneity and compatibility between different datasets can be challenging [27]. Despite advances in cloud and parallel computing, the computational demands of extrapolating forest attributes across large scales remain challenging, prompting the need to evaluate the efficacy of time series data compared to single-year predictors in mapping new forest attributes.

As the interest in utilizing time series data for forest inventory updates grows, there remains a notable gap in research regarding the comparative effectiveness of single-year and time series predictors when it comes to extrapolating vertical forest complexity. Surprisingly, forest complexity assessment comprises a mere 3% of the literature exploring modeling LiDAR-derived estimates of forest attributes [28]. However, the study of horizontal structure has been significantly advanced by integrating remote sensing indices, texture analysis, and ground-truth measurements [29–31].

Texture analysis, a technique used to discern spatial variations in pixel values [32], elevates our interpretation of high-resolution satellite and aerial imagery. It enables the identification of patterns and structures within the forest landscape, thereby enhancing our understanding of forest spatial heterogeneity. Consequently, the need for evaluating whether established methods that capture spatial heterogeneity in horizontal forest structure can effectively be applied to extrapolated vertical forest structure becomes imperative and timely, especially with the increasing prevalence of LiDAR extrapolation data integrated with satellite imagery [33]. Therefore, it is vital to explore the application of texture analysis and its potential to bridge the gap between vertical and horizontal complexity.

Given the escalating global ecological concerns, the focus of our study is to determine the most effective methodologies for constructing comprehensive maps of forest complexity, leveraging both time series and single-year Landsat predictors. Our geographic area of interest is the northern boreal forest of North America, a region of paramount importance for its diverse ecological dynamics and the accelerated rate at which it is undergoing climate change [34–36]. This rapid ecological transformation intensifies the forest biome's susceptibility to disturbances such as wildfires, pest outbreaks, and extreme weather events [37]. Recognizing the pressing need for enhanced understanding of the ecosystem's responses to these changes, initiatives like the Arctic–Boreal Vulnerability Experiment (ABoVE) have been launched [38–40]. As part of this initiative in particular, data collected in 2019 by the Land, Vegetation, and Ice Sensor (LVIS) across the vast regions of Alaska, Yukon, and the Northwest Territories (AYNWT) provide an unparalleled resource to develop comprehensive maps of vertical and horizontal forest structure.

In this study, we ask the following: (1) Do predictive models of vertical forest structure improve when single-year Landsat metrics are replaced with time series metrics in the

northern boreal forest? (2) When using time series metrics, what length of time yields the best results for forest complexity estimates—15-year or 30-year predictors? (3) Is it possible to use a moving window on the vertical complexity to calculate horizontal complexity? (4) What statistical metrics and window sizes are most appropriate for characterizing forest horizontal complexity? Answers to these questions provide valuable insights into the trade-offs between the accuracy of forest complexity estimates with time series versus single-year data. Our findings have important implications for forest management, conservation efforts, and further development of remote sensing-based forest inventory methods.

## 2. Materials and Methods

### 2.1. Study Area

The study encompasses the northern part of the North American boreal forest, including Alaska, Yukon, and the Northwest Territories (AYNWT) (Figure 1). This region is defined by a mosaic of coniferous and deciduous trees, with species such as spruce (*Picea mariana* and *Picea glauca*), trembling aspen (*Populus tremuloides*), and balsam poplar (*Populus balsamifera*) on the landscape. The forest understory includes shrubs like willows (*Salix* spp.), birch (*Betula* spp.), and alders (*Alnus* spp.), as well as herbaceous plants.

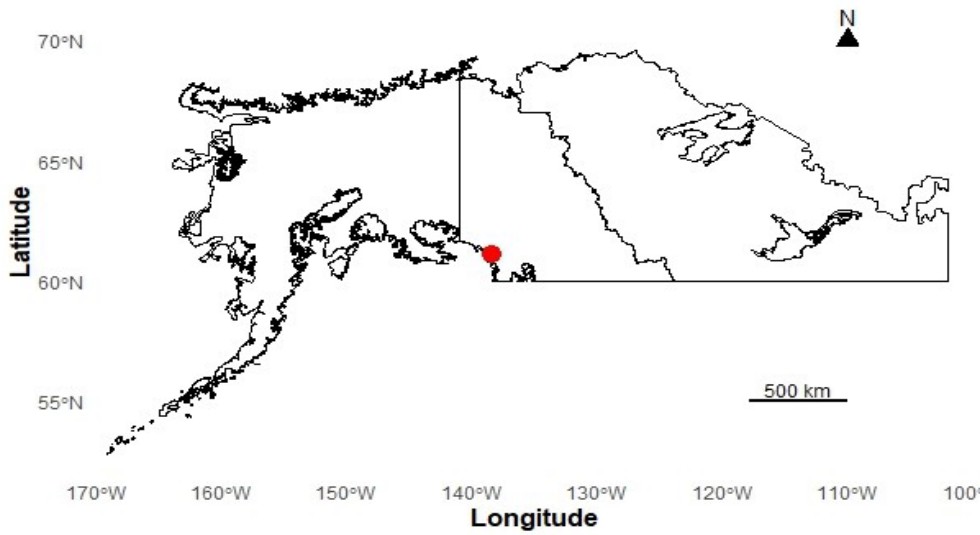

**Figure 1.** Boreal forest within the regions of Alaska (USA), Yukon, and the Northwest Territories (Canada). The red point indicates the location of the Kluane Valley in southwestern Yukon. The boreal forest data are derived from the research of Brandt [41].

Boreal forests in this region exhibit a heterogeneous stand structure with variations in tree density, age, and height. The forest's structure is influenced by environmental factors like climate, soil, topography, and natural or anthropogenic disturbance events [42,43]. Across this expansive landscape, the tree density fluctuates from sparse woodlands to dense forest stands [44]. This region experiences long, cold winters and short, cool summers, with winter precipitation primarily falling as snow. The widespread presence of permafrost is also responsible for distinct soil and vegetation features [45].

Our analysis of the horizontal complexity is primarily concentrated in the Kluane region of southwest Yukon (61°07′N 138°24′W) (Figure 1). In this region, the forest canopy includes mostly white spruce and aspen, while the understory is predominantly occupied by willows and birch [46].

### 2.2. Datasets

#### 2.2.1. LVIS-L3 Full-Waveform LiDAR Data

The ABoVE (Arctic–Boreal Vulnerability Experiment) LVIS (Land, Vegetation, and Ice Sensor) L3 Gridded Vegetation Structure dataset represents the vegetation structure across

North America (2017–2019), and is available through the Oak Ridge National Laboratory Distributed Active Archive Center (ORNL DAAC) https://daac.ornl.gov/ABOVE/guides/ABoVE_LVIS_VegetationStructure.html (accessed on 1 August 2023). The L3 dataset comprises gridded information on the canopy relative height (RH), complexity, canopy cover (CC), ground elevation, and the number of LVIS footprints used to generate each pixel's estimate, at 30 m resolution. The dataset provides information on vertical vegetative structure using relative canopy height metrics supplemented by canopy cover estimates at various heights. The data covers a diverse range of landscapes, in support of the Arctic–Boreal Vulnerability Experiment (ABoVE) and Global Ecosystem Dynamics Investigation (GEDI)-related scientific objectives. For our study, we selected the 2019 data of the AYNWT, as it offers the most recent dataset. Then, we clipped these data to the boundaries of the North American boreal forest extent [41].

### 2.2.2. Landsat

We used Landsat data derived from the Landsat Surface Reflectance Tier 1, including Landsat 5™, Landsat 7 (ETM+), and the Landsat 8 (OLI) sensors at 30 m spatial resolution, and with spectral bands from visible to short-wave infrared and a 16-day revisit time (with single sensors) [47]. Landsat surface reflectance data are generated by applying atmospheric corrections to raw satellite images, removing the effects of atmospheric scattering and absorption, and providing surface representation that facilitates analysis of land cover and other environmental variables [48]. Landsat Surface Reflectance Tier 1 is subject to strict quality control for geometric and radiometric criteria. We harmonized the spectral consistency among sensors in the Landsat archive (Landsat 5, 7, and 8 satellites) by adopting appropriate regression slope and intercept values from Roy et al. [27].

### 2.3. Data Processing

#### 2.3.1. Estimating Vertical Forest Complexity

Understanding changes in forest ecosystems needs reliable and consistent data across time and space. We selected summer median composite images (1989–2019) to ensure temporal alignment with LVIS data collection, eliminating any temporal mismatches. This approach was chosen, as it effectively eliminated outliers and provided a well-balanced representation of forest conditions [49]. In addition, this approach provided us with statistical consistency across time and sensors, an essential feature for enhancing the validity of our multi-temporal study.

We generated a random sample of LVIS L3 complexity data using approximately 70,000 points separated by >500 m to ensure a diverse representation of the study area and reduce the likelihood of sampling bias [50] (Figure 2). Complexity is a comprehensive metric derived from the waveform captured by the LVIS sensor. It factors in both the count and intensity of peaks within a waveform, acting as an indicator that highlights the deviation of the waveform from a uniform surface. In forested areas, this waveform complexity mirrors the vertical complexity of the vegetation structure [51].

The training dataset included relevant predictor variables at the locations of the random sample points, including data for a single year (2019) and a time series of Tasseled-cap Greenness (TCG), Tasseled-cap Brightness (TCB), and Tasseled-cap Wetness (TCW) constructed from the Landsat composites [52]. We included elevation via ALOS DSM v3.3 to obtain cross-boundary (USA and Canada) consistency at 30 m. Additionally, we calculated slope, and latitude and longitude were used to account for location-specific variation where geographic differences correspond to environmental and climatic variations influencing forest structure. Coordinates aid in distinguishing between areas with similar vegetation index values but different forest structures. For the time series (15 and 30 years), we estimated long-term means, standard deviation, and slope of the tasseled cap indices. We used Sen's Slope because it is a non-parametric technique offering advantages of being robust to outliers, lacking normality, and heteroscedasticity [53].

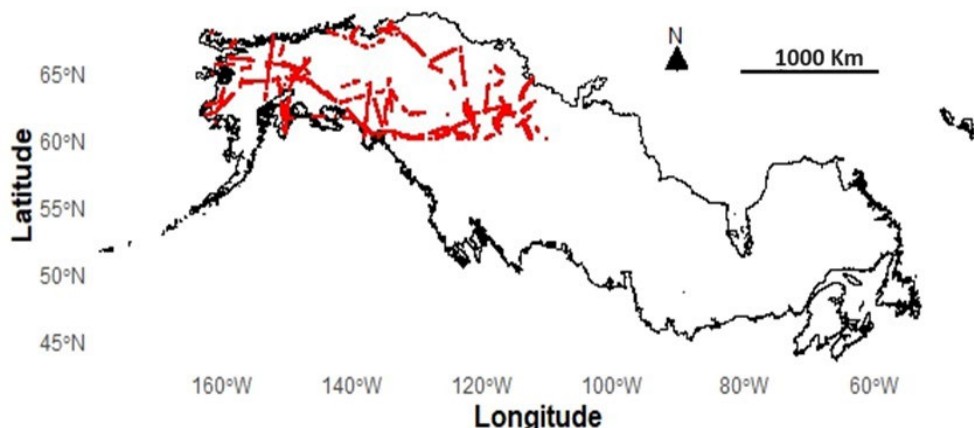

**Figure 2.** Boreal forest of North America derived from Brandt [41] showing the approximately 70,000 sampling points of the study region in red.

For unbiased modeling, we used the same sampling points for all 5 random forest regression models predicting vertical forest complexity, incorporating elevation, slope, latitude, and longitude as covariates (Table 1). While the single-year model included the 2019 tasseled cap values, the other models integrated time series tasseled cap metrics, including mean, standard deviation, and Sens Slope calculated from either 15 or 30 years of data [20]. Two models were designed to assimilate both time series and single-year metrics. Random forest models were trained with 70% of the data and computed with 10 trees, with all other parameters set to default (variables per split = square root the number of predictors; bag fraction = 0.5; min leaf population = 1; max nodes = null) [54]. Accordingly, we tested each model's accuracy with the remaining 30% of the data. The coefficient of determination ($R^2$) and relative root mean squared error (RRMSE) were used to test model accuracy between observed and predicted values, as well as model performance. To identify any significant differences in model residuals, we employed the Kruskal–Wallis test on model residuals by limiting validation data to the same 5000 points.

**Table 1.** Predictor variables for each model. The filled boxes represent the variables used in each model.

| | Model A: Single-Year Predictors | Model B: 15-Year Predictors | Model C: Single-Year and 15-Year Predictors | Model D: 30-Year Predictors | Model E: Single-Year and 30-Year Predictors |
|---|---|---|---|---|---|
| **Single-year:** 2019 TCB, TCG, TCW | ▓ | | ▓ | | ▓ |
| **Time series:** 2004–2019: Mean, Standard Deviation, Regression Slope of TCB, TCG, TCW | | ▓ | ▓ | | |
| 1989–2019: Mean, Standard Deviation, Regression Slope of TCB, TCG, TCW | | | | ▓ | ▓ |
| **Location:** Latitude and Longitdue | ▓ | ▓ | ▓ | ▓ | ▓ |
| **Topographic:** Elevation, Slope | ▓ | ▓ | ▓ | ▓ | ▓ |

To bolster the performance and generalizability of our single-year model, we conducted a model calibration [55]. Simulations involving 10–50 trees and a bag fraction of 0.5–0.7 to gauge model accuracy vis-à-vis the time series approach. We refrained from modifying other parameters, as we observed a notable increase in $R^2$ and a decrease in RRMSE. By employing the single-year model (Model A), we extended forest complexity estimates to regions where only response variables were available, owing to its comparable performance with other models.

2.3.2. Estimating Horizontal Forest Complexity

We used the single-year model and texture measurements to convert vertical forest complexity to horizontal complexity. This texture analysis was performed on a moving window that encompassed both first-order and second-order statistics [56], thus capturing the spatial variability within the forest landscape.

We applied a spatial filter to calculate a metric of spatial variability within the moving window, creating a continuous surface of horizontal complexity. We evaluated three window sizes (3 × 3, 5 × 5, and 7 × 7) to balance spatial resolution and smoothing effect, initially using a 5 × 5 window to estimate entropy, variance (first-order), and GLCM variance (second-order) metrics, with the most intuitive and simple metric prioritized as a descriptor of horizontal forest complexity [57].

To identify redundancy among texture metrics, we conducted a Spearman rank correlation test [57]. We validated our approach by comparing our optimized texture metric with a traditional horizontal structure assessment technique, summarizing pixel values with mean and standard deviation from texture computations within a 100 m radius for each sample point [57,58]. We treated the mean and standard deviation as separate measures, one representing central tendency and the other data dispersion, and compared them across all sample points using a Spearman correlation.

To illustrate our approach, we selected the Kluane region, southwest Yukon to characterize effectiveness of the moving window methodology in estimating horizontal forest complexity. To substantiate our results, we evaluated the explanatory power of our derived horizontal complexity layer in explaining densities of snowshoe hare (*Lepus americanus*) pellets across the landscape; hare pellet counts are correlated with hare population density [59–61]. The premise of this effort is that hares tend to occupy and select habitats that have abundant horizontal cover [62–64].

## 3. Results

### 3.1. Vertical Forest Complexity

Random forest models revealed a strong correspondence between predicted and observed values, with the 30-year time series model including single-year predictors (Model E) exhibiting the highest accuracy ($R^2$ = 0.84; RRMSE = 8.97%). Generally, longer time series metrics improved model accuracy ($R^2$ = 0.77–0.84; RRMSE = 9.0–10.3%), though the 30-year model (Model D) was an exception, with only a small difference in accuracy ($R^2$ = 0.8; RRMSE = 9.96%). Furthermore, incorporating single-year predictors into both the 15- and 30-year time series models enhanced their predictive power and accuracy. Specifically, in Model B, there was a slight $R^2$ increase from 0.81 to 0.83 coupled with an RRMSE descent from 9.91% to 9% in Model C. Correspondingly, in Model D, an increase in $R^2$ was observed from 0.8 to 0.84, along with a decline in RRMSE from 9.96% to 8.97% in Model E (Table 2).

**Table 2.** $R^2$ and RRMSE values between observed and predicted values for the validation data using the same validation points.

| Metric | Model A: Single-Year Predictors | Model B: 15-Year Predictors | Model C: Single-Year and 15-Year Predictors | Model D: 30-Year Predictors | Model E: Single-Year and 30-Year Predictors |
|---|---|---|---|---|---|
| $R^2$ | 0.77 | 0.81 | 0.83 | 0.8 | 0.84 |
| RRMSE | 10.30% | 9.99% | 9% | 996% | 897% |

Despite improvements in model accuracy with additional predictors and extending the time series, all of the models maintained a comparable level of agreement between the observed and predicted values, indicating their practical utility and effectiveness. Scatterplots in Figure 3 show improved agreement from Model A to Model E. Upon a closer examination of the residual values with a Kruskal–Wallis test reveals no statistically

significant differences between the residuals of both models ($p > 0.9$). Given its accuracy and comparable performance to more complex models, Model A was chosen for projecting vertical forest complexity across the AYNWT region (Figure 4).

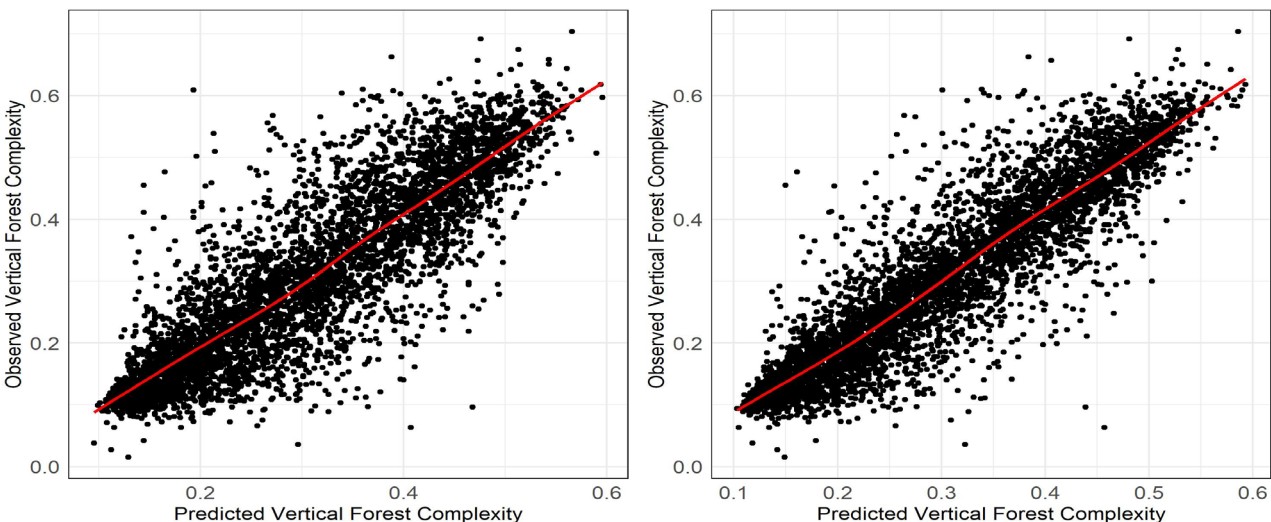

**Figure 3.** Scatterplots between observed and predicted attributes for the validation data using single-year (Model A on the left) and 30-year plus single-year Landsat predictors (Model E on the right). Observed values are LVIS L3 complexity. The red line is the linear trend.

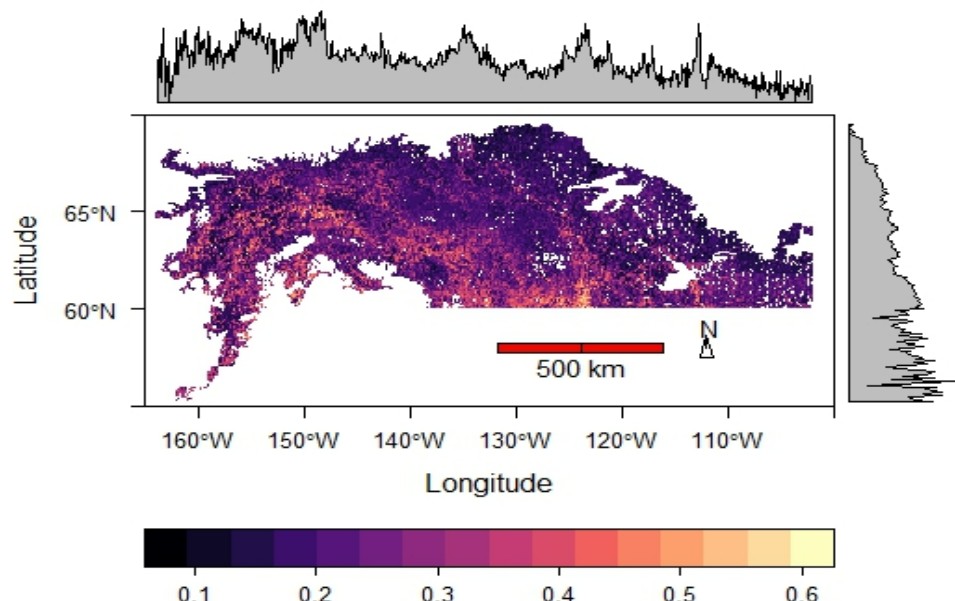

**Figure 4.** Spatial predictions of vertical forest complexity across the boreal regions of Alaska, Yukon, and the Northwest Territories. The color scale shows the predicted values from less to most complex. Adjacent marginal plots showcase the average forest complexity values distributed across latitude and longitude. Note: All white areas signify either bodies of water or regions with unavailable data.

Post-calibrated Model A exhibited enhanced precision over the initial single-year model (Figure 5), denoted by a rise in R2 from 0.77 to 0.84 and a decline in RRMSE from 10.3% to 8.4%. Significantly disparate residuals ($p < 0.001$) between the pre-tuned and calibrated Model A underscored the potential of a single-year predictor to outperform time series models with basic parameter adjustments. The variable importance for our best-performing models can be found in Appendix A.

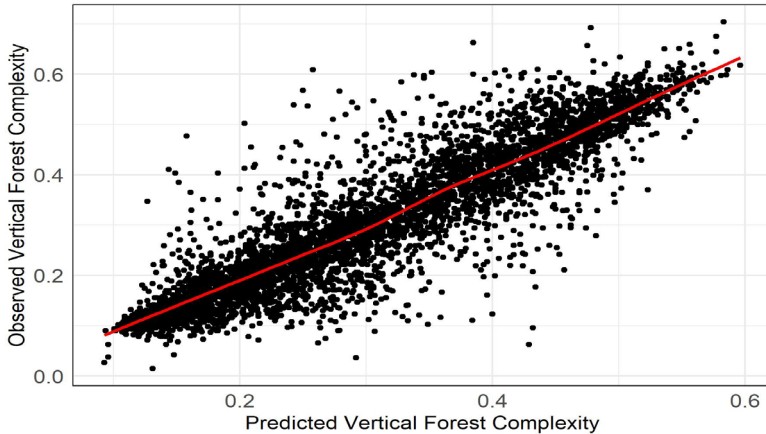

**Figure 5.** Scatterplot between observed and predicted attributes for the validation data using hyperparameter tuning on the single-year model. The red line is the linear trend.

### 3.2. Horizontal Forest Complexity

We clipped the vertical forest extrapolation layer for the Kluane Valley region. Figure 6 presents the results of the 5 × 5 moving estimated metrics, along with their corresponding Spearman's correlations. Among the metrics, variance displayed robust correlations of rs = 0.91 and rs = 0.77 with entropy and GLCM variance, respectively. These strong correlations, coupled with inherent simplicity, highlight the merits of sample variance as an effective metric for assessing horizontal forest complexity in this study. The relationship between the mean value of the texture-estimated variance and standard deviation of the original vertical complexity layer was strong (R2 = 0.84; RRMSE = 8.4%), illustrating the effectiveness of using texture analysis with a 5 × 5 window to estimate horizontal forest complexity.

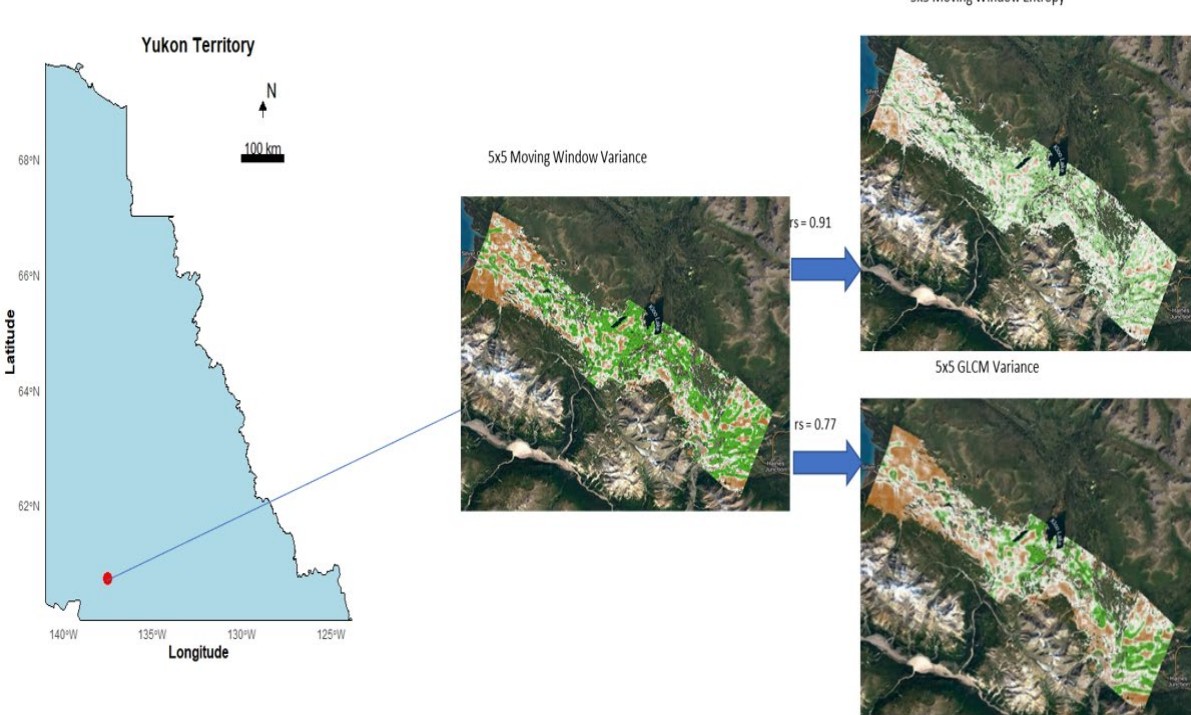

**Figure 6.** Horizontal forest complexity for the Kluane Valley, Yukon, using 5 × 5 moving window estimated metrics and their respective Spearman's correlations. Brown = Low, White = Medium, Green = High horizontal complexity. The figure illustrates the strong correlation of variance with entropy and GLCM variance.

We applied the same procedure using both 3 × 3 and 7 × 7 windows, and the results consistently aligned with those from the 5 × 5 window. Specifically, the estimated variance from the 3 × 3 window demonstrated a Spearman's correlation coefficient 0.86, versus 0.90 for the 7 × 7 window. Notably, the 5 × 5 window exhibited an equivalent correlation coefficient of 0.9. These findings underscore the effectiveness of the 5 × 5 window in capturing the essence of horizontal forest structure, all while preserving intricacies of fine-scale variability. As such, it is evident that the 5 × 5 window is well-suited for estimating the horizontal forest structure from pre-existing vertical forest complexity data. The estimated variance with the 3 × 3 window presented a rs value of 0.86, slightly lower than the 0.9 shown by both the 7 × 7 and 5 × 5 windows.

## 4. Discussion

### 4.1. Vertical Forest Complexity

Vegetation structure complexity strongly influences animal habitat selection and ecosystem processes, but measuring this nuanced heterogeneity at broad scales is challenging, especially in the boreal forests of Alaska, Yukon, and the Northwest Territories. These forests face pressures from climate change, forest fires, pest outbreaks, and human activities, which modify forest structure and habitats. Thus, accurately characterizing vegetation structure in these regions is vital for mapping, monitoring changes, and informing proactive conservation strategies in dynamic environmental conditions.

Emerging methodologies that leverage LiDAR-derived forest attributes, Landsat predictors, and terrain metrics present a potent solution, enabling the estimation of crucial structural forest attributes even in regions without existing LiDAR data [65,66]. Our analysis supports these findings by revealing strong relationships between predicted and observed values, aligning with the findings of recent research [21]. While most of the research has primarily concentrated on forest biometric and stand attributes such as forest height, DBH, basal area, and stem volume, as opposed to vertical or horizontal complexity, our findings stand out [67,68]. The results indicate a superior model fit than most research, with strikingly accurate RRMSE and R2 values of 8.97% and 0.84, respectively, and compare favorably with analogous research that employs full-waveform LiDAR for estimating vertical structure [69].

Previous studies have shown that Landsat time series are more effective at capturing variability in structural forest attributes than single-year Landsat predictors [28]. While our research aligns with these findings, we did not observe the same substantial increase in accuracy reported in other studies. As an example, Pflugmacher et al. [69] reported that Landsat time series significantly improved model predictions over single-year predictors for four forest structural attributes (e.g., above ground live biomass: R2 = 0.80; RRMSE = 41% versus R2 = 0.58; RRMSE = 57%). Bolton et al. [20] documented a significant increase in accuracy (e.g., forest top height: R2 = 0.25; RRMSE = 22.8% to R2 = 0.54; RRMSE = 16.5%) when transitioning from single-year to 30-year Landsat predictors, while our findings show marginal improvement (vertical complexity: R2 = 0.77; RRMSE = 10.3% to R2 = 0.8; RRMSE = 9.96%) for the same transition in the time series. Similarly, Bolton et al. [21] found that longer Landsat time series (>15 years) consistently provide more accurate estimations of forest structural attributes across different forest types, productivities, and histories of disturbances (R2 = 0.45–0.70). In contrast, we observed that extending the time series from 15 to 30 years did not improve the accuracy (R2 = 0.81 to R2 = 0.80), whereas adding single-year predictors to both time series lengths provided a greater accuracy boost. This contrasts with Bolton et al. [20], where including single-year predictors slightly reduced the accuracy for certain attributes within the 30-year time series (e.g., net stem volume: R2 = 0.57; RRMSE = 34.1% to R2 = 0.55: RRMSE = 34.9%). Consequently, we find no significant benefits in developing time series predictors for extrapolating vertical forest structure complexity.

While time series predictors exhibited slight improvements in model accuracy compared to the single-year model, these benefits are marginal. Additionally, our residual

analysis showed no significant differences between residuals in the single-year and most comprehensive time series model, highlighting this point further ($p > 0.9$). Also, a considerable increase in predictive accuracy is observed when shifting from a 30-year model (R2 = 0.80; RRMSE = 9.96%) to a calibrated single-year model (R2 = 0.84; RRMSE = 8.97%), as revealed by pronounced differences in residuals between pre-tuned and calibrated models ($p < 0.001$). Therefore, given the challenges associated with processing time series data, time series predictors may not always be necessary or beneficial for extrapolating vertical forest complexity. However, our findings are specific to the northern boreal forest of North America and spatial extrapolations. For temporal predictions, using time series predictors could be valuable by capturing dynamic ecosystem changes over time, including seasonality, long-term trends, forest growth, disturbance recovery, and stressor responses [70,71]. Notably, to our knowledge, there are no existing direct comparisons utilizing LiDAR measurements as the response variable and Landsat time series as predictors for estimating vertical forest complexity. Thus, our study represents an initial attempt to understand the successful employment of Landsat time series predictors and terrain attributes for this purpose.

Our approach diverges from other studies in that we employ median composites rather than the commonly used best available pixel (BAP) composite approach or annual time series stacks of images [22,70]. In contrast to image stacks, they provide a single, annually representative image of the forest. These traits are especially beneficial for time series analyses, as they facilitate detection of actual trends and reveal underlying patterns in the imagery. While median composites potentially hold advantages over other compositing methods in the realm of time series predictors in the boreal forest [72], this claim requires additional research to validate, particularly when comparing them to BAP composites.

Our study adopts a distinct sampling methodology compared to many others. While many favor a stratified sampling approach for LiDAR data, valuing its capability to comprehensively sample all features and its efficiency in reducing spatial autocorrelation, we chose a systematic random sampling method [50]. This method provided extensive coverage over all strata due to the large volume of data collected. While we acknowledge that this approach may not be as efficient and could lean towards capturing more abundant features in the landscape, our results show that the accuracy remained consistent across the entire range of complexity values. This reaffirms our confidence in the effectiveness of our sampling approach, which successfully encompassing the full spectrum of variability within our study area.

We focused on capturing forest structure variability across a vast region with a large sample size. This heterogeneity likely enhanced the training model and its generalizability, resulting in improved model accuracy compared to studies with a more localized focus. However, we did not account for disturbances directly, which may have introduced bias, as Landsat time series imagery has shown good predictive capability for structural attributes in recently disturbed areas [20,70]. Disturbance history has also aided in modeling biomass and other attributes over a 30-year period [22]. Including disturbances as predictors could have improved the accuracy of our time series models, but it was beyond the scope of our study.

Although other studies had comparable sample sizes and more extensive spatial coverage than ours [19], their extrapolated forest attributes did not achieve the accuracy levels of ours (R2: 0.12–0.61 and RRMSE: 24.5–78.7% versus our R2 = 0.84 and RRMSE = 8.4% in our single-year model calibration). However, comparing the accuracy metrics of our study with those of others requires careful interpretation, since other research primarily focused on point cloud-derived forest attributes, which may interact differently with Landsat-derived spectral predictors and spectral saturation based on canopy closure [33].

Despite using fewer predictor variables (no disturbance history or other spectral indices) than other similar studies [20,21,70] we achieved superior agreement between predicted and observed values. This may be largely attributed to the successful integration of vertical forest complexity measurements from large footprint full-waveform systems

with Landsat imagery and terrain attributes, as seen with other forest attributes [69,73–76]. Moreover, preliminary analysis in areas without recent disturbances, our models have consistently maintained the relationships between predictor variables. This robustness in maintaining these relationships can be attributed to the limited changes observed in forest structures over preceding years, demonstrating the robustness of our modeling approach in regions with relatively low disturbance rates.

### 4.2. Horizontal Forest Complexity

Texture analysis has consistently demonstrated its ability to capture vegetation structure [77–79]. In this research, we found that vertical complexity can be effectively converted into a measure of horizontal complexity using suitable texture analysis methods. Among the metrics considered, variance particularly stands out in assessing horizontal forest complexity, with robust correlations with entropy and GLCM variance (rs = 0.91 and rs = 0.77, respectively) and a strong association with the standard deviation of original vertical complexity ($R2 = 0.84$; RRMSE = 8.4%).

Previous studies support our findings. For example, Hudak and Wessman [79] and Wood et al. [57] found that first-order standard deviation and variance, respectively, were effective in measuring vegetation structure in different ecosystems. However, our study approach limits direct comparisons, and our results diverge from studies like Kayitakire et al. [80], which propose variance as being less effective for capturing specific forest structural attributes. This suggests that researchers may need to tailor metric selection to the specific forest attributes under study.

Applying a $5 \times 5$ window to estimate variance proved highly effective in capturing horizontal forest structure at our study location, showing a robust correlation (rs = 0.90) with an adapted method of measuring horizontal vegetation structure [59]. Although using $3 \times 3$ and $7 \times 7$ windows resulted in consistent outcomes, the $3 \times 3$ window exhibited a slightly lower correlation coefficient (rs = 0.86), suggesting less optimal performance in capturing horizontal forest structure compared to the other windows. However, the $5 \times 5$ window outperforms the $7 \times 7$ window in preserving fine-scale phenomena that are crucial for delineating complex forest structure. Larger windows may smooth out details, reducing sensitivity to finer variations of vertical complexity. For localized analysis of horizontal forest complexity, the $5 \times 5$ window, paired with variance, offers an advantageous approach.

In a separate ongoing analysis, we are further validating our horizontal complexity layer by examining its relation to snowshoe hare pellet densities. Preliminary findings from our investigation reveal that horizontal complexity emerged as the second most influential variable, after vertical complexity, in estimating hare pellet densities in the Kluane Valley (Appendix A). This aligns with previous studies that established a strong relationship between dense horizontal vegetation structure and habitat use and density of snowshoe hares [81–84], and further highlights the ecological relevance of robustly estimating forest complexity.

This study proposes an innovative approach for quantifying horizontal complexity derived from extrapolated vertical complexity and provides initial evidence of its ecological significance. For future research, we recommend tailoring window size to the study's objectives: smaller windows for analyzing horizontal structure in heterogeneous landscapes, and larger windows for quantifying horizontal structure at a landscape scale. For our study, the $5 \times 5$ window effectively captured horizontal structure, but different window sizes may be more appropriate depending on the scale of analysis.

## 5. Conclusions

Our research highlights the effectiveness of methodologies incorporating full-waveform LiDAR, Landsat predictors, and terrain metrics to accurately characterize forest structural complexity. The study finds superior models fit with calibrated single-year models over 15- and 30-year time series, questioning the often-preferred time series approach for pre-

dicting forest attributes. Nonetheless, these findings are specific to the northern boreal forest of North America, and future research is warranted to validate these methodologies in other forest ecosystems, at different spatial scales, and with other forest attributes. Furthermore, building upon existing research, our study suggests that texture analysis can effectively convert vertical vegetation complexity into horizontal complexity, with variance standing out as an important metric. Our innovative approach deviates from traditional methods, utilizing LiDAR extrapolations and a moving window to perform the conversion. This method, while new, offers valuable insights, and preliminary results also indicate its ecological relevance, as shown by its correspondence with snowshoe hare pellet densities. Nonetheless, we urge future studies to tailor their window size selection to their specific objectives.

**Author Contributions:** Both authors collaborated to conceptualizing the study. N.D.-K. conducted the analysis, generated the figures and tables, and initially drafted the manuscript, with revisions and contributions from D.L.M. Both authors actively participated in the study's development and jointly approved its submission. All authors have read and agreed to the published version of the manuscript.

**Funding:** This research was funded by a Collaborative Special Projects grant from the Natural Sciences and Engineering Research Council of Canada (NSERC), and the writeup was supported by the Canada Research Chairs Program.

**Data Availability Statement:** L3 LVIS data can be accessed through the Oak Ridge National Laboratory, (ORNLDAAC) https://daac.ornl.gov/ABOVE/guides/ABoVE_LVIS_VegetationStructure.html (accessed on 1 August 2023). Landsat data can be downloaded and accessed through Google Earth Engine, https://developers.google.com/earth-engine/datasets/catalog/landsat (accessed on 1 August 2023).

**Conflicts of Interest:** The authors declare no conflict of interest.

**Appendix A**

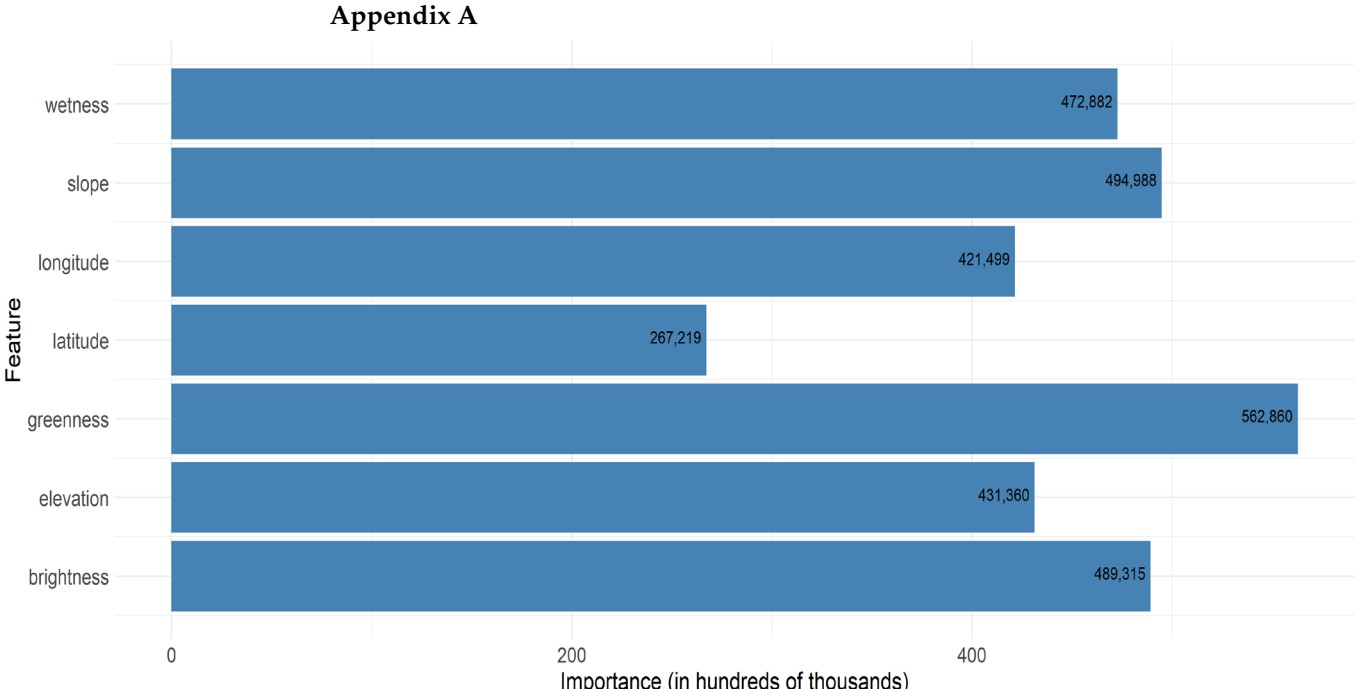

**Figure A1.** Random forest variable importance for the calibrated single-year model (Model E). This bar plot illustrates the importance of each feature in the best performing model.

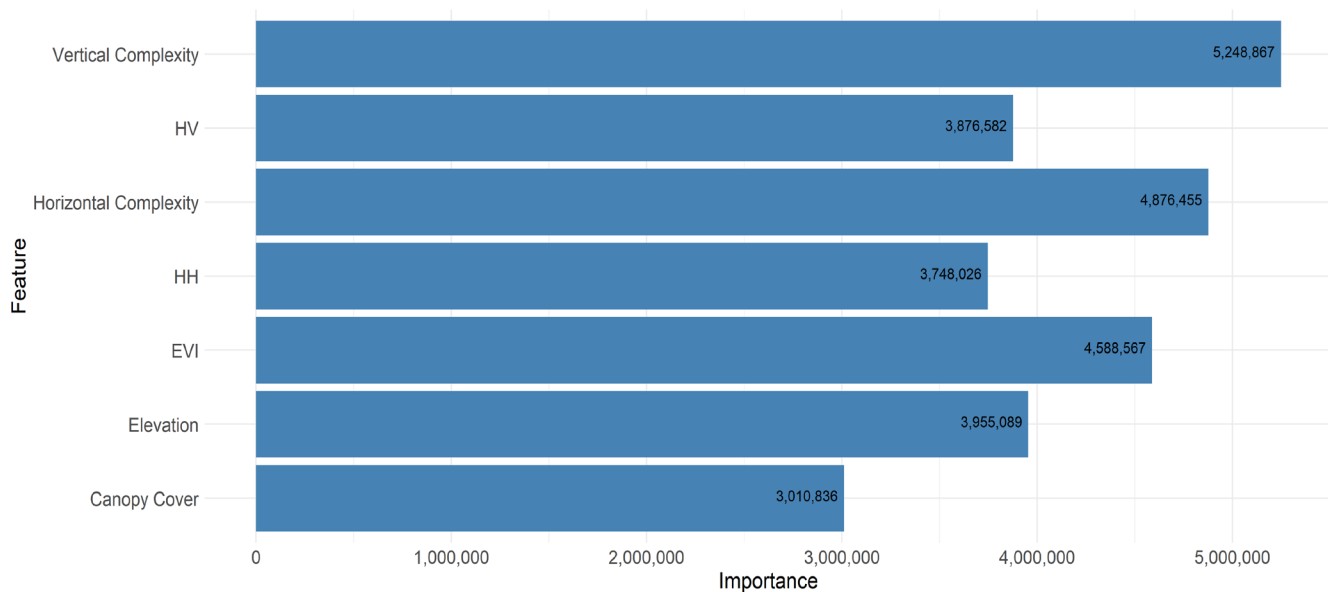

**Figure A2.** Shows the importance of variables in a random forest regression for predicting snowshoe hare pellet densities in the Kluane landscape. The analysis incorporates metrics of vertical and horizontal complexity derived from this study. Additional variables include elevation, EVI (enhanced vegetation index), SAR polarizations (HH and HV), and canopy cover.

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
