# Peer review of "Bridging the Gap: Comprehensive Boreal Forest Complexity Mapping through LVIS Full-Waveform LiDAR, Single-Year and Time Series Landsat Imagery"

_remotesensing, doi:10.3390/rs15225274_

Round 1
Reviewer 1 Report
Comments and Suggestions for Authors
General comments
This paper addresses one of the currently most important areas of the globe where direct observations are difficult and remote sensing is very useful if not obligatory.
In general, the manuscript is well written and easy to follow. There are formatting issues that obscured some of the data. I had few specific comments.
The goal is estimating forest complexity but I don’t see a precise definition of complexity. It seems essentially a measure of differences in sensed textures. The justification seems the assumed connection between diversity and ecosystem resilience can be extended to complexity of observed textures. It would be nice if the work could be tied to more tangible need; for example, how does this relate to permafrost melt. The snowshoe hare example in appendix is not legible and is not discussed in either results or discussion. I missed any reference to the appendix.
Specific comments
Line 118 long sentence would be more readable to break as …………height. and Forest ………
Line 279 -286 there is a formatting problem of text running into the table.
Line 322 – another format problem only half of figure is visible.
Line 364-379 There seems to be no caption for this figure I assume is figure 6
Author Response
Thank you for your comments and helpful feedback. We agree that a more detailed description of vertical complexity, specifically measured by LVIS should be included in the manuscript. We have included a few lines in the data processing section. Although we agree that forest complexity could be linked to issues like permafrost melt, we are unaware of the tangible evidence. To our knowledge, the primary drivers of permafrost loss include climate warming, changes in snow dynamics, and increased frequency of wildfires. Also, geophysical attributes, such as slope, play a pivotal role in permafrost in Alaska , but vegetation complexity seems to be severely understudied in this regard.
Regarding the snowshoe hare example in the appendix, we acknowledge the oversight, but we have addressed it in the discussion section and referenced the appendix as well.
Thank you for the comments regarding sentence structure and formatting issues, they have been solved in the manuscript.
Reviewer 2 Report
Comments and Suggestions for Authors
I have read with great pleasure the article titled "Bridging the Gap: Comprehensive Boreal Forest Complexity Mapping through Full-waveform LiDAR, Single-year and Time Series Landsat Imagery". The study is aimed at determining the most effective methodologies for constructing comprehensive maps of forest complexity, leveraging both time-series and single-year Landsat predictors, answering questions related to the improvement of predictive models of vertical forest structure, best parameters of time-series metrics, appropriateness of using a moving window, and best fitted statistical metrics and window sizes. The study presented in the manuscript is scientifically sound and mature, and the research deep enough. Moreover, the authors demonstrate the novelty of their study, proposing an innovative approach deviating from traditional methods, utilizing LiDAR extrapolations and a moving window to perform the conversion, which offers valuable insights, and has ecological relevance. They also demonstrate that their study makes a significant contribution to the theoretical advancement of the field, through the need for evaluating whether established methods that capture spatial heterogeneity in horizontal forest structure can effectively be applied to extrapolated vertical forest structure. The topic is original and relevant to the field, addressing a specific gap in the field regarding the comparative effectiveness of single-year and time-series predictors when it comes to extrapolating vertical forest complexity, which is correctly identified by the authors. The findings make an addition to the subject area compared with other published materials, providing valuable insights into the trade-offs between accuracy of forest complexity estimates with time-series versus single-year data; this contribution is also correctly identified by the authors. The methodology is complex, well presented, and scientifically correct. The manuscript is of potential interest to a broad international audience, well written, documented by appropriate and numerous references and supported by appropriate tables and figures, and the results are relevant for the methodological advancement of the field, with important implications for forest management, conservation efforts, and further development of remote sensing-based forest inventory methods. The conclusions are sound and appropriate for a broad international audience. The manuscript deserves to be published by "Remote Sensing". I have a minor comment, aimed at improving the presentation.
The article would certainly benefit upon expanding the discussions to include possible limitations of their methodology, and elaborate a little bit more on the future research directions, beyond what is presented in lines 509-513 and 529-530. In addition, there are some minor issues, such as: a title ending with a period, an image that cannot be seen completely on page 7, parentheses open but not closed (line 66), a table overlapped with the text (page 6); all these do not diminish the value of their study, but require a careful check of their manuscript.
Author Response
Thank you for your feedback. We apologize for the oversight regarding the formatting issues. The uploading process may have altered the original formatting, but we will ensure all mentioned discrepancies are rectified in the revised manuscript. We also added an additional paragraph regarding our sampling approach and how it may lead to inaccuracies and lack of efficiency, which is perhaps one of the biggest problems with our methods.
Reviewer 3 Report
Comments and Suggestions for Authors
The authors presented and interesting research regarding inferencing forest complexity (vertical and horizontal) based on Landsat imagery, DSM and location. For the purposes of reference and training the openly accessible LiDAR-based product was used. To inference the vertical complexity, the authors tested random forest algorithm with different sets of features derived from Landsat imagery. All the models included coordinates and topographic information.
While the conducted experiment is interesting, sometimes it was hard to understand the context because of the formatting issues. Some of the images are not visible, some are half visible, sometimes the text overlaps with the table. It needs to be fixed before the second round of the review so that all the results can be properly interpreted.
I have 5 major issues that I believe should be better explained in the paper:
1) All the models include coordinates as features used by random forest to predict the complexity of forest vertical structure. It is a valid assumption that the forest vertical structure is similar in the points that are close to each other. However, this limits the applicability of the model to the selected test area. What is more, if the training pixels were selected randomly, we can expect that each testing pixel has some of the training pixels close to it. This may heavily affect the results. What happens when the model is used to predict vertical structure for the areas outside of the test area (and far away from training pixels)? Could you explain why the coordinates should be used for this study?
2) As I understood, the summer season Landsat imagery was used for training and testing the model. However, there is no information when the LiDAR data was collected (the LiDAR data is usually collected for a leaf-off season). How this discrepancy affects random forest predictions? Was any difference observed between the accuracy achieved for areas with deciduous and coniferous trees?
3) As a consequence of the above, the LiDAR data should be described more thoroughly – more information about what features are available and which of them are actually used, the acquisition season, etc.
4) Was the feature selection performed for any of the models? The multi-year models use substantially more features than single-year models. The more features used the bigger probability of overfitting the model. Could it be the reason why the single-year model gives similar accuracy as multi-year models?
5) It would be interesting to see if the results achieved for the year 2019 are confirmed when performing the analysis for other year.
Smaller issues:
1) The formatting needs to be heavily corrected. The references have too many brackets. Images are not visible; tables interfere with text. Sometimes it makes it impossible to properly analyze the results.
2) The abstract (and title) should make it clear that the LiDAR is not directly used for the investigation, but the ready-to-go product is used.
3) In the Figure 1 and 2, it would be beneficial to see where the forests are in regard to the training/testing pixels. Is the whole area forested?
4) Questions 3 and 4 are too detailed for the introduction. It is hard to understand the context without reading the whole paper first. Please, try to make the more general.
5) It would be nice to see how all the models perform when they are “calibrated”, not only one of them.
6) The horizontal forest complexity is not mentioned as a feature for LVIS data. Where was the reference taken from?
7) The images lack labels, for instance: what does a color mean in figure 4?
Author Response
Thank you for the extensive feedback and comments.
1) All the models include coordinates as features used by random forest to predict the complexity of forest vertical structure. It is a valid assumption that the forest vertical structure is similar in the points that are close to each other. However, this limits the applicability of the model to the selected test area. What is more, if the training pixels were selected randomly, we can expect that each testing pixel has some of the training pixels close to it. This may heavily affect the results. What happens when the model is used to predict vertical structure for the areas outside of the test area (and far away from training pixels)? Could you explain why the coordinates should be used for this study?
Its true that proximity in geographical coordinates can often imply similarity in forest vertical structure. However, in our study, we've taken measures to minimize the influence of spatial autocorrelation. Even if pixels might be situated closely, we've ensured a minimum distance of 500 meters between them to diminish this effect.
The rationale for using latitude and longitude has two reasons. Firstly, the northern boreal forest, covering Yukon, Alaska, and Northwest Territories is a very large region. The extent of this spatial scale means that variations in geographic coordinates correspond with differences in the local environment, history of climatic conditions, and, consequently, forest structure. By including longitude and latitude in our model, we account for such location-specific variations that significantly influence forest attributes.
Moreover, vegetation indices might be similar in values across the entire boreal forest. Thus, without a specific spatial context, partitioning variance in the model would become more challenging. By integrating geographic coordinates, we're adding a layer of spatial differentiation that aids the model in distinguishing between areas with similar vegetation index values but different forest structures.
Also, it's essential to recognize that any variable we employ, be it geographical coordinates or different indices, would inherently be more applicable to the locations with data. This fitting issue is a recurring challenge in many interpolations or extrapolation modeling. To address this and test the model's generality, we set aside 30% of the data for validation. Given that some of these validation points were distant from the training sites, the results we obtained are indicative of a model with strong generalizability.
2) As I understood, the summer season Landsat imagery was used for training and testing the model. However, there is no information when the LiDAR data was collected (the LiDAR data is usually collected for a leaf-off season). How this discrepancy affects random forest predictions? Was any difference observed between the accuracy achieved for areas with deciduous and coniferous trees?
The LVIS data collection took place during the summer, eliminating concerns about temporal inconsistencies. We added a few lines regarding these issues the “Estimating Vertical Forest Complexity” Section. While it would be intriguing to assess accuracy variations between areas with deciduous versus coniferous trees, it demands highly precise landcover maps for the entire region. Based on our experience, widely used options, such as those derived from MODIS, have proven unreliable in certain areas we examined.
3) As a consequence of the above, the LiDAR data should be described more thoroughly – more information about what features are available and which of them are used, the acquisition season, etc.
We agree it has been fixed within the manuscript.
4) Was the feature selection performed for any of the models? The multi-year models use substantially more features than single-year models. The more features used the bigger probability of overfitting the model. Could it be the reason why the single-year model gives similar accuracy as multi-year models?
We did not perform explicit feature selection. Instead, we carefully chose relevant variables based on past research. This method ensured that our model was informed by relevant predictor variables. We opted for the Random Forest regression primarily because of its less prone to overfitting. This choice is supported by studies such as those conducted by Bolton et al. (2018 & 2020). In their research, the use of multiple features in time-series models led to increased accuracy and generalizability over single year models, suggesting that this should not be an issue.
5) It would be interesting to see if the results achieved for the year 2019 are confirmed when performing the analysis for other year.
We are currently undertaking these back-casting exercise in more localized regions of the boreal forest. This research serves as a foundational step for those efforts. We included a few lines regarding this issue in the discussion.
Smaller issues:
1) The formatting needs to be heavily corrected. The references have too many brackets. Images are not visible; tables interfere with text. Sometimes it makes it impossible to properly analyze the results.
Yes, the formatting keeps getting messed up when we upload it, but this will be taken care of.
2) The abstract (and title) should make it clear that the LiDAR is not directly used for the investigation, but the ready-to-go product is used.
We included the fact that the LiDAR data came from LVIS in both the title and the abstract.
3) In the Figure 1 and 2, it would be beneficial to see where the forests are in regard to the training/testing pixels. Is the whole area forested?
Our sampling approach was intentionally designed to encompass a wide range of variability within the northern boreal forest ecosystem, which includes not only forests but also shrublands and potentially recently disturbed areas up to 2019. This approach was chosen to provide a more comprehensive representation of the entire biome. Additionally, our validation results demonstrate that our sampling strategy effectively captured the variability present in the study area. We have also added a discussion section that addresses this issue more explicitly.
4) Questions 3 and 4 are too detailed for the introduction. It is hard to understand the context without reading the whole paper first. Please, try to make the more general.
To address this concern, we have some revisions to the text to hopefully offer a more general perspective.
5) It would be nice to see how all the models perform when they are “calibrated”, not only one of them.
We appreciate your suggestion, and it's a valid point. However, the primary objective of this paper is to compare single-year and time-series models using consistent parameters. We wanted to demonstrate that even without extensive calibration, the single-year model can outperform time-series models. That said, we acknowledge that further calibration of all models could provide valuable insights, and we agree that this could be a logical next step in future research.
6) The horizontal forest complexity is not mentioned as a feature for LVIS data. Where was the reference taken from?
Yes, its not a feature from LVIS, we refer to converting vertical complexity into horizontal complexity with texture analysis.
7) The images lack labels, for instance: what does a color mean in figure 4?
Thank you for pointing that out, we have included more information on the figure descriptions.
Round 2
Reviewer 3 Report
Comments and Suggestions for Authors
Thank you for your answers. I have two more suggestions regarding question 1.
1. I suggest the authors include in the paper the reasoning behind choosing coordinates for the variables. Also, the part of the answer mentioning distant pixels present in the validation data should be included in the paper because we cannot see the separation into training and validation in Figure 2.
2. Could you include in the paper the analysis of the model variables importance that is provided by Random Forest algorithm? I think that it would give the readers more insight into how the model operates.
Author Response
Thank you for your helpful comments and suggestions!
We included a more detailed response regarding the use of latitude and longitude in the data processing section.
In response to the previous round, I think there might have been a little more clarification needed in terms of pixel distance. While some training and validation locations may be close, they are no closer than 500 meters apart. The linear regression models demonstrate that there is no bias at any specific range of values, indicating the model's effective capture of forest structures across a wide range of complexities with the validation data (no matter if close or far from the training data). Regarding figure 2, distinguishing training and validation pixels using different colors, especially with red representing approximately 70,000 points, leads to a cluttered and non-interpretable visualization. At this scale, color overlaps occur, making individual sampling points challenging to interpret.
For the sake of comprehensiveness and clarity, we included a figure breaking down the feature importance for our best-performing model, Model E (in the appendix). This inclusion aims to provide readers with a tangible understanding of which variables significantly influence our best model's outcomes.
Thank you once again for your constructive comments and suggestions. We hope the manuscript is ready for publication.